# Super Critical Fluid Extracted Fatty Acids from *Withania somnifera* Seeds Repair Psoriasis-Like Skin Lesions and Attenuate Pro-Inflammatory Cytokines (TNF-α and IL-6) Release

**DOI:** 10.3390/biom10020185

**Published:** 2020-01-25

**Authors:** Acharya Balkrishna, Pradeep Nain, Anshul Chauhan, Niti Sharma, Abhishek Gupta, Ravikant Ranjan, Anurag Varshney

**Affiliations:** 1Drug Discovery and Development Division, Patanjali Research Institute, Haridwar 249 405, India; 2Department of Allied and Applied Sciences, University of Patanjali, Patanjali Yogpeeth, Haridwar 249 405, India

**Keywords:** fatty acids, *Withania somnifera* seeds, SCFE, GC-FID, inflammation, TPA-induced psoriasis, cytokines, reactive nitrogen species, NFκB, IL-6, TNF-α

## Abstract

(1) Background: *Withania somnifera* Dunal (Ashwagandha) is a widely used medicinal herb in traditional medicinal systems with extensive research on various plant parts. Surprisingly, seeds of *W. somnifera* have never been investigated for their therapeutic potential. (2) Methods: *W. somnifera* seeds were extracted for fatty acids (WSSO) using super critical fluid extraction, and was analyzed by gas chromatography. Its therapeutic potential in psoriasis-like skin etiologies was investigated using a 12-O tetradecanoyl phorbol 13-acetate (TPA)-induced psoriatic mouse model. Psoriatic inflammation along with psoriatic lesions and histopathological scores were recorded. WSSO was also tested on murine macrophage (RAW264.7), human epidermoid (A431), and monocytic (THP-1) cells, stimulated with TPA or lipo poly-saccharide (LPS) to induce pro-inflammatory cytokine (IL-6 and TNF-α) release. NFκB promoter activity was also measured by luciferase reporter assay. (3) Results: Topical application of WSSO with concurrent oral doses significantly reduced inflammation-induced edema, and repaired psoriatic lesions and associated histopathological scores. Inhibition of pro-inflammatory cytokines release was observed in WSSO-treated A431 and THP-1 cells, along with reduced NFκB expression. WSSO also inhibited reactive nitrogen species (RNS) in LPS-stimulated RAW264.7 cells. (4) Conclusion: Here we show that the fatty acids from *W. somnifera* seeds have strong anti-inflammatory properties, along with remarkable therapeutic potential on psoriasis-like skin etiologies.

## 1. Introduction

*Withania somnifera* Dunal (WS) is also known as Indian Ginseng or Ashwagandha [1] or the wonder plant of the Indian medicinal system. It is a multipurpose medicinal plant rich in a host of pharmaceutically active ingredients [2], highly valued in traditional Ayurvedic [3], Unani [4], and Chinese [5] medicinal systems. Chemical analysis of roots and leaves of the WS plant has resulted in the identification of active compounds called withanolides, which are anti-inflammatory molecules and known to modulate NFκB activity [6,7]. The other known bioactive ingredient of the WS plant, withanolide sulphoxide, inhibits COX-2 expression [8], whereas withaferine-A displays anti-cancer and anti-tumor activity by modulating various signaling pathways [9] as well as having neuroprotective activities [10]. WS plant extracts are mainly prepared from roots [11,12] and leaves [13,14], which have been studied extensively. Alcoholic extraction of plant parts results in the collection of essential oil, which is different from fixed seed oils [15]. Interestingly, no biological study has been performed on *W. somnifera* seed components, although chemical analysis of chloroform-methanol extracted oil [16] and the steroidal constituents of seeds have been investigated recently [17]. We have extracted fatty acid rich contents from WS seeds (WSSO) by the supercritical fluid extraction (SCFE) method. SCFE is a “green” extraction technology that employs liquid CO_2_ under high pressure. The resultant WSSO is oily in nature and can be easily applied topically on skin [18]; therefore, investigating its therapeutic potential against skin diseases such as psoriasis [19] is a natural first choice of biological study for WSSO.

Psoriasis is a chronic inflammatory, recurrent skin disease affecting a range of populations in different countries, from 0.09% to 11.4%, almost 125 million of the world’s population [20]. It is characterized by marked hyper-proliferation and modified differentiation of the dermis, resulting in scaly erythematous plaques located at skin exteriors. Psoriasis is understood to be genetically predisposed [21] and stimulated by external stimuli including stress, infection, environmental factors, or drugs [22]. Psoriasis is an autoimmune inflammatory disease, the result of unregulated crosstalk between immune cells [21], especially inflammatory cells such as macrophages, T cells, and epidermal keratinocytes. Pathophysiological abnormalities include hyperkeratosis and parakeratosis [22] in skin, vascular hyperplasia, infiltrations of T lymphocytes [23], among others. After decades of studies and research, the treatment of psoriasis is still based on controlling the flaring symptoms and inflammation using topical [24] and systemic therapies [25]. Topical therapies used are corticosteroids [24] and retinoids [26], whereas systemic medicines include common first-line drugs like methotrexate [27] and cyclosporin [28]. Apart from these, many anti-inflammatory medicines are used both topically and systemically to reduce the inflammation. Prolonged use of these pharmaceutical drugs, which are basically either immunosuppressant or anti-inflammatory in nature [29], causes various other unwarranted complications [30].

Natural origin products are known to have minimal side effects even after continuous use [31] and easily available locally. *Withania somnifera* Dunal is one such herb. Our study is aimed at analyzing the anti-psoriatic potential of its seed fatty acids, WSSO. These fatty acids were extracted using supercritical CO_2_ fluid [32] and analyzed by gas chromatography. Chemical analysis of WSSO reveals the presence of various fatty acid components, known to repair damaged skin [33]. Anti-psoriatic studies were performed on a 12-O tetradecanoyl phorbol 13-acetate (TPA)-induced psoriasis-like mouse model. The diseased animals were treated with both topical skin application and systemic oral administration of WSSO. Effects of WSSO on the reduction of TPA-induced psoriatic edema, epidermal thickness, hyperkeratosis, and skin histopathological lesion scores were measured. Biochemical analysis of skin biopsy revealed an inhibition in myeloperoxidase (MPO) activity with WSSO treatments. In order to analyze the underlying mechanism of action, various cell-based in vitro experiments were performed using A431 (human epidermoid), THP-1 (human macrophage), and RAW264.7 (mouse macrophage) cell lines. Effects of WSSO on TPA or lipo poly-saccharide (LPS)-induced effects on psoriatic or inflammation markers were investigated by measuring the release of pro-inflammatory cytokines, IL-6 and TNF-α, in epidermal origin A431 and macrophage THP-1 cells. NFκB is the central transcription factor involved in the pathogenesis of psoriasis [34] and inflammation [35]. WSSO treatment stabilized the stimulated NFκB expression in THP-1 cells. WSSO also reduced the reactive nitrogen species (RNS) in RAW264.7 cells, indicating reduced synthesis of nitric oxide, a potent modulator of inflammation and psoriasis. Taken together, we provide conclusive evidence that WSSO indeed possesses therapeutic value, inhibits inflammation, and reduces TPA-induced psoriatic-like symptoms both in in vivo and in vitro studies. 

## 2. Materials and Methods

### 2.1. Chemicals and Reagents

Cell culture media for in vitro culture, RPMI1640, DMEM, fetal bovine serum (FBS), penicillin-streptomycin mixture were obtained from Thermo Fisher (Waltham, MA, USA). Fine reagents for biological studies such as LPS (cat#L2630), 3-(4,5-dimethylthiazol-2-yl)-2,5-diphenyltetrazolium bromide (MTT), 3,3′,5,5′-Tetramethylbenzidine (TMB), Griess reagent, phorbol-12-myristate-13-acetate (PMA) (cat#P1585), TPA, indomethacin and dexamethasone (DEXA) were purchased from Sigma-Aldrich (St. Louis, MO, USA). HPLC standard withaferine A/B was obtained from Sigma Aldrich, whereas withanolide A and withanoside IV/V were procured from Natural Remedies Pvt. Ltd., (Bengaluru, India). ELISA kits for cytokines interleukin-6 (IL-6) and tumor necrosis factor (TNF-α) were purchased from BD Biosciences (San Jose, CA, USA). Boron trifluoride (BF_3_) reagents, n-heptane, hematoxylin, mercury (II) oxide red and potassium aluminum sulphate dodecahydrate were procured from Merck (Mumbai, Maharashtra, India). Eosin yellow and FeCl_3_ were obtained from Hi-Media Laboratories (Mumbai, Maharashtra, India). All the other chemicals and reagents used in the studies were of the highest commercial grade.

### 2.2. Isolation of WSSO from W. somnifera Seeds

*W. somnifera* (WS) seeds were provided by Patanjali Gramodhyog Trust, Haridwar, India (Batch number D4/RM/0096/0719). The supercritical carbon dioxide (SC-CO_2_) extraction was performed in a supercritical extractor SFE 5000 Bio-Botanical Extraction System with a single separator equipped with a CO_2_ recycler (Waters Corporation, Milford, MA, USA). WS seeds were pulverized, powdered and mixed with food-grade liquid CO_2_. Backflow pressure was regulated at 450/80 and 55 ± 5/40 Bar/°C, with CO_2_ flow rate at 60.0 g/min for 22 h 20 min. Static extraction was followed by dynamic extraction.

### 2.3. Measurement of Fatty Acid Contents in WSSO

A gas chromatography-flame ionized detector (GC-FID) was used to determine the fatty acid content in WSSO by following the American Chemist’s Society method [36]. Briefly, SCFE extract of WSSO was mixed with 0.5 N methanolic NaOH and heated to create fat globules. After 10 min, BF_3_-methanol was added to the mixture, still under boiling condition, and further kept for 2 min. After the addition of n-Heptane, it was boiled for 1 min. The whole mixture was then removed from the heat and a saturated NaCl solution was added and mixed vigorously. After keeping on standby for 10 min at room temperature, the top layer of n-Heptane containing fatty acid mixture called fatty acid methyl ester (FAME) was collected. Using helium as a carrier gas, GC-FID analysis was performed on a GC-2010 Plus (Shimadzu, Kyoto, Japan) gas chromatograph. C4-C24 FAME standard mix, containing a neat mixture of 37 FAMEs, was purchased from Sigma Aldrich (Product code 18919-1AMP) and was used to analyze WSSO contents.

### 2.4. HPLC Analysis of WSSO, and Post SCFE Exhaust of WS Seeds

WSSO (0.5 g) and the post-SCFE exhaust of WS seeds (0.25 g) were diluted with 5 mL of methanol and sonicated for 30 min. This solution was centrifuged for 5 min at 8000 rpm and filtered through 0.45 µm nylon filter paper. For the post-SCFE exhaust of WS seeds, the solution was further diluted 5 times with the same solvent. Quantification of marker compounds was performed on an HPLC system (Waters Corporation, USA) consisting of a binary pump (1525), a detector PDAD (2998) and an auto-sampler (2707). A reverse-phase column Shodex C18-4E (5 µm, 4.6 × 250 mm) was used and the column temperature was maintained at 27 °C. HPLC mobile phase A (0.14 g KH_2_PO_4_; dissolved in 1000 mL water, containing 0.5 mL H_3_PO_4_) and mobile phase B (acetonitrile) at a flow rate of 1.5 mL/min were used with the defined gradient program. For each sample analysis, 20 µL of the test solution was injected, and chromatograms at 227 nm were recorded. 

### 2.5. In Vivo Studies for Anti-Inflammatory and Anti-Psoriasis Potential

#### 2.5.1. Experimental Animals

Male C57bl/6 mice of bodyweight 18–22 g were procured from Charles River licensed animal supplier, Hylasco Biotechnology Pvt. Ltd., Hyderabad, India. All the animals were placed under a standard controlled environment of 12:12 h light and dark cycle. Mice were provided with sterile filtered water ad libitum and a standard pellet diet (Purina Lab Diet, St. Louis, MO, USA; product code 5L79). Animal experimental protocols were performed under study number LAF/PCY/2019-022, duly approved by the Institutional Animal Ethics Committee (IAEC), and were conducted in accordance with animal ethics guidelines.

#### 2.5.2. Generation of TPA-Induced Psoriasis-Like Mouse Model

The TPA-induced mouse ear skin inflammation model, as described previously, was employed with minor modifications [37,38]. Briefly, 20 μL of TPA solution (125 μg/mL in acetone) was applied topically on the mice ear on days 0, 2, 4, 6, 8, and 10. For vehicle control, the ear was treated with 20 μL of acetone alone on the same days. Ear thickness was measured every day using a digital vernier caliper (Mitutoyo, Tokyo, Japan). Increase in ear thickness was determined by subtracting the ear thickness of day 0 (before TPA or acetone application) from the respective time point thickness. Anti-psoriatic activity of WSSO was investigated on animals stimulated with TPA. Treatment regime was followed as described for other ayurvedic oil based formulations for skin disorders, where topical application on affected skin was complemented with an oral dose supplement [38,39,40]. Animals were treated with 20 μL topical application WSSO along with 410 or 1230 mg/kg p.o. or with DEXA at 0.2 mg/ear (topically) throughout the experiment, as indicated in Figure 1. WSSO animal oral doses were calculated following Jacob and Nair’s method [41], based on estimated human daily dose of 2 to 6 g/day, for other comparable oil based formulations [42]. The anti-psoriasis activity (%) was calculated for each animal on day 10 (D10), using the following formula: [Mean Ear Edema of TPA Control mice − Ear Edema of each Mouse of Test or DEXA treated Mouse]/[Mean Ear Edema of TPA Control mice] × 100

#### 2.5.3. Histopathological Analysis

Animals were humanely sacrificed on day 10 after 6 h of the last drug treatment. Ear biopsy samples were weighed and fixed in 10% (*v*/*v*) neutral buffered formalin, embedded in paraffin, and sectioned at 3–5 μm. The sections were then stained with hematoxylin and eosin (H&E). The thickness of the epidermis (from the basal layer to stratum corneum) was measured by MagVision image analysis software using a Magcam DC5 microscopic camera with calibration by a stage micrometer. Blinded histopathological analysis was performed to reduce biasness in observing the severity of the lesions. These were labeled as NAD = no abnormality detected, 1 = minimal (<1%), 2 = mild (1–25%), 3 = moderate (26–50%), 4 = moderately severe/marked (51–75%), 5 = severe (76–100%) and distribution of the lesions were recorded as focal, multifocal, and diffuse. Other parameters viz. extent of the lesion, severity of hyperkeratosis, number and size of pustules, epidermal hyperplasia (measured in interfollicular epidermis), severity of inflammation in the dermis and soft tissue, and any other lesion(s) were taken into consideration in the histopathological examination and scoring. Histopathological lesion score was computed for the individual animal, as the sum of each pathological score. These were then averaged for a given study group, with standard error estimation to generate total lesion scores.

#### 2.5.4. MPO Assay

Myeloperoxidase activity using 3,3′,5,5′-tetramethylbenzidine (TMB) was measured, as described [43]. Briefly, 10 µL of ear biopsy homogenized samples were combined with 80 µL 0.75 mM H_2_O_2_ and 110 µL TMB solution (2.9 mM TMB in 14.5% DMSO and 150 mM sodium phosphate buffer at pH 5.4), and the plate was incubated at 37 °C for 5 min. The reaction was stopped by the addition of 2 M H_2_SO_4_. The absorption was measured at 450 nm to estimate MPO activity using the Envision microplate reader (PerkinElmer, Shelton, CT, USA).

### 2.6. In Vitro Experiments to Study Anti-Inflammatory Potential of WSSO

#### 2.6.1. In Vitro Cell Culture

A431, THP-1, and RAW264.7 cell lines were obtained from the ATCC licensed repository National Centre for Cell Science (NCCS), Pune, India and cultured in RPMI-1640 media and DMEM media containing sodium pyruvate (1 mM) and L-glutamine (4 mM), as per requirements. Media was supplemented with 10% heat-inactivated fetal bovine serum (FBS), penicillin-streptomycin (100 U/mL). All the cells were grown at 37°C in a sterile humidified 5% CO_2_ incubator and experiments were performed in the sterile environment. Monocyte THP-1 cells were differentiated into macrophages by treatment of PMA (50 ng/mL) for 48 h as described, with minor adjustments [44], resulting in adherence of differentiated macrophages. These adherent and differentiated THP-1 cells were used for all the subsequent experiments.

#### 2.6.2. Cell Viability Study

WSSO was prepared as an emulsion in complete culture media (RPMI-1640) after brief sonication in water bath. THP-1 cells were seeded in a 96-well plate at the density of 10,000 cells per well in a flat bottom 96-well plate. The cells were incubated with the WSSO at the concentrations of 1.25, 2.5, 5, 10, 15, and 30 μL/mL for a period of 24 h. At the end of the exposure time, consumed media was aspirated and 100 μL incomplete media containing 0.5 mg/mL tetrazolium dye (MTT) was added to each well and incubated for 3 h at 37 °C. To terminate the experiment, an equal amount of the consumed media was carefully removed, and an equal amount of DMSO was added in all the wells to dissolve the formazan crystals. Absorbance was recorded at 570 nm using the Envision microplate reader (Perkin Elmer, USA). Cell viability percentage was calculated, using negative control (untreated cells) as 100% viable cells.

#### 2.6.3. Reactive Nitrogen Species (RNS) Measurement

RAW264.7 cells were seeded in 96-well culture plates at a density of 2 × 10^5^ cells/mL. Cells were treated with different concentrations of WSSO fatty acid emulsion, made in incomplete DMEM media and incubated for 1 h. Cells were then stimulated with LPS (500 ng/mL) and incubated for an additional 18 h at 37 °C in a CO_2_ incubator. The RNS release in the culture media was determined using modified Griess reagent (Sigma), following the manufacturer’s protocol. Absorbance was recorded at 540 nm using the Envision microplate reader (Perkin Elmer, USA), and percent change in total RNS release was measured.

#### 2.6.4. Cytokines Level Measurement

A431 or THP-1 cells were seeded in 24-well culture plates at a density of 5 × 10^5^ cells/mL. For the experiment, WSSO was prepared as a fatty acid emulsion and mixed with the cell culture media at different concentrations: 0.1, 0.3, 1, 3, and 10 μL/mL. THP-1 or A431 cells were pre-incubated with the WSSO containing media for 1 h before the addition of 500 ng/mL (final concentration) LPS or 10 µg/mL TPA, except in the negative control. The cell culture supernatants were harvested after 24 h to measure different pro-inflammatory cytokines (IL-6 and TNF-α) using ELISA kits (BD Biosciences, USA) following the manufacturer’s protocol. Absorbance was recorded at 450 nm using the Envision microplate reader (Perkin Elmer, USA).

#### 2.6.5. Luciferase Reporter NFκB Expression Assay

Luciferase reporter vector with NFκB promoter sequence upstream of the luciferase gene was transiently transfected into adherent and differentiated THP-1 cells. Transfection was performed using the Lipofectamine 3000 (Invitrogen, Grand Island, NY, USA), following the manufacturer’s instruction. The experiment was performed as described earlier [45], with minor modifications. The consumed media was carefully aspirated 48 h post-transfection, and fresh media was added. The experimental wells were treated with different concentrations of WSSO. After 1 h, LPS was added at a final concentration of 500 ng/mL where required and incubated further for 12 h. D-Luciferin salt (Perkin Elmer, USA), at a final concentration of 150 μg/mL, was added to the cells and incubated at 37 °C in the dark. Relative percentage changes in luminance intensity were measured from each well using the Envision microplate reader (Perkin Elmer, USA). Measurements from LPS alone wells were considered as 100% activity of the NFκB reporter gene.

### 2.7. Statistical Analysis

The data are expressed as mean ± standard error of means (SEM) for each group. Statistical analysis was performed using GraphPad Prism software version 7.03. Two-way analysis of variance (ANOVA) followed by the Newman–Keuls multiple comparison test was used to calculate the statistical difference in absolute ear edema. A one-way ANOVA followed by Dunnett’s multiple comparison post-hoc test was used to calculate the statistical difference in cytokine(s) analysis, ear biopsy weights, epidermal thickness, and lesion scores. The values of *p* < 0.05 were considered statistically significant, unless stated otherwise.

## 3. Results

### 3.1. SCFE Extraction of WS Seeds by CO_2_ and Analysis of Fatty Acid Contents by GC-FID

A total 1800 g of WS seed powder was subjected to the supercritical CO_2_ extraction procedure. After optimizing extraction protocol, 234.0 g of oily material (WSSO) was obtained as the final yield. The percentage extracted yield was calculated as ~13%. For GC-FID analysis, WSSO was mixed with different solvents to obtain fatty acid methyl esters (FAME). The FAME from WSSO was isolated and investigated using GC-FID. The detailed analysis of WSSO confirmed presence of four major peaks and couple of minor peaks (Figure 2). Different fatty acids were identified and quantified by peak analysis with respect to reference standards (Table 1). These fatty acid components were identified and individually quantified as following: linoleic acid (54.15%), oleic acid (23.16%), palmitic acids (12.64%), stearic acid (4.07%), 11,14,17-eicosatrienoic acid (0.66%), and nervonic acid (0.38%). A few unidentified fatty acid components were also present in the WSSO and found to be below the limit of quantification of the GC-FID detector (Table 1).

### 3.2. WSSO Contains Trace Amount of Withanolides

HPLC analysis of WSSO and the post-SCFE exhaust of WS seeds was performed (Figure 3) and compared with the known standard mix of withanolides. The majority of withanolides, signature contents of Withania, were detected in the post-SCFE exhaust of WS seeds and represented almost total withanolide content in WS seed. The WSSO contains only 0.022% *w*/*w* withaferine A and 0.001% *w*/*w* withanolide A, as compared to 0.807% *w*/*w* withanoside IV, 0.359% *w*/*w* withaferine A, and 0.744% *w*/*w* withanoside V in the post-SCFE exhaust of WS seeds. Minor peaks of withanolide A (at retention time, 38.65 min) and withaferine A (at retention time, 31.86 min) in the WSSO chromatogram is negligible in comparison to WS seed exhaust (Figure 3). Withanolide B and withanone were not detected in any of the test samples.

### 3.3. In Vivo Anti-Psoriatic Activity of WSSO

#### 3.3.1. WSSO Inhibits Ear Edema in TPA Induced Psoriatic Mouse

After topical application of TPA to the mouse ear (2.5 µg/ear) for induction of psoriasis-like disease, WSSO treatment was given by topical application of 20 µL and concurrent oral doses (410 mg/kg or 1230 mg/kg). TPA caused a progressive increase in ear edema in the diseased (TPA CON) animals as treatment continued to day 10 (Figure 4A). Highly significant changes (*p* < 0.001) in ear edema were observed in diseased animals compared to negative control (NC) animals. Anti-inflammatory drug DEXA was applied topically to psoriatic ear (0.2 mg/ear) in the standard control group, causing significant (*p* < 0.001) reduction in the ear edema from day 2 onwards (*p* < 0.001) (Figure 4A). In the WSSO-treated group, both concurrent oral (410 and 1230 mg/Kg) and topical application (20 µL/ear) of WSSO caused a reduction in ear edema from day 5 onwards. Both of the WSSO doses caused a decrease in edema; however, at a higher dose of 1230 mg/kg, reduction of ear edema was more significant (*p* < 0.001). 

Till day 7, no significant variation was observed in ear edema after treatment with different doses of 410 and 1230 mg/kg dose of WSSO (with 20 µL of T.A.). However, on days 8, 9, and 10, significant inhibition of ear edema (*p* < 0.001) was observed compared to disease control (TPA CON) at the higher dose of 1230 mg/kg (Figure 4A,B). The percent inhibition in ear edema was calculated on day 9 of DEXA and WSSO 410 and 1230 mg/kg treated mice and calculated to be 41.29% ± 1.90%, 18.09% ± 4.90% and 28.27% ± 4.04%, respectively, in comparison to TPA CON animals (Figure 4B).

#### 3.3.2. WSSO Reduces Psoriatic Ear Epidermal Thickness and Weight

Ear biopsies were obtained after the termination of the experiment on D-10 (Figure 1). Continuous TPA stimulation for ten days induced severe inflammatory response with a significant (*p* < 0.001) increase in ear biopsy weight in disease control animals (TPA CON) (Figure 5A). The psoriatic animals treated with DEXA had a significant reduction in the ear biopsy weight (*p* < 0.001) compared to disease control animals (Figure 5A) with ~60% inhibition in ear biopsy weight (Figure 5B). In WSSO-treated groups, topical application of 20 μL along with different oral doses of 410 and 1230 mg/Kg decreased ear biopsy weights. The decrease in ear biopsy weight was significant (*p* < 0.001) at the higher dose of 1230 mg/kg (Figure 5A), as compared to TPA CON animals. The inhibition of inflammation-induced biopsy weight was dose-dependent in the WSSO-treated psoriatic mice (Figure 5B). Histopathological analysis of psoriatic ear punch biopsies also demonstrated that psoriatic animals (TPA CON) displayed significant (*p* < 0.001) increases in epidermal thickness (85.48 ± 10.05 μm) (Figure 5C) compared to negative control animals (11.00 ± 1.65 μm). 

In vehicle control animals (12.10 ± 0.80 μm) where only acetone was used, no significant changes were observed compared to NC. In TPA-induced psoriatic ear, topical application of DEXA significantly (*p* < 0.001) reduced the epidermal layer thickness (37.3 ± 8.07 μm) compared to TPA CON. In the WSSO-treated group, significant reduction (*p* < 0.001) in epidermal thickness was observed in a dose-dependent manner. Topical application complemented with oral treatment of 410 and 1230 mg/kg caused a reduction in epidermal thickness to 61.10 ± 15.61 and 34.92 ± 6.40 μm, respectively (Figure 5C). Topical application of DEXA also significantly (*p* < 0.001) reduced epidermal layer thickness (37.30 ± 8.07 μm), as expected.

#### 3.3.3. WSSO Reduces MPO Activity in Ear Biopsy Tissue

Myleoperoxidase (MPO) activity was measured in ear punch biopsies post-termination of the experiment (Figure 1 and Figure 5D). Vehicle-treated control animals (VC) have insignificant changes in MPO activity compared to negative control animals (NC). On the other hand, TPA-induced animals displayed significant (*p* < 0.001) increase in MPO activity (2.86 ± 0.48 μM/mg) compared to NC animals (0.79 ± 0.47 μM/mg). Topical application of WSSO concurrent with 430 and 1230 mg/kg oral doses reduced MPO activity to 2.03 ± 1.24 and 1.29 ± 1.05 μM/mg, respectively. However, these changes were significant (*p* < 0.05) in 1230 mg/kg treated animals compared to TPA CON. DEXA (0.2 mg/ear T.A.) also significantly (*p* < 0.001) inhibits MPO activity to 1.05 ± 0.83 μM/mg (Figure 5D).

#### 3.3.4. WSSO Repairs Psoriatic-Like Skin Lesions

Histopathological analysis of ear biopsies revealed detailed structural changes after stimulation with TPA, causing hyperkeratosis, hyperplastic epidermis, pustule and rete ridge formation. Infiltration of the inflammatory cells was also observed in the dermal region (Figure 6C). In NC animals, skin was normal (Figure 6A), and similarly, no changes were observed in vehicle control (VC) animals (Figure 6B). DEXA treatment to TPA-induced psoriatic skin reduced the number of inflammatory cells and pustule formations (Figure 6D), with almost unchanged hyperkeratosis and hyper-plasticity in the epidermis. Surprisingly, the concurrent oral and topical application of WSSO on the psoriatic mice ear reduced signs of hyperkeratosis and hyper-plasticity (unlike DEXA) in the skin epidermis; however, inflammatory cells were still found in the dermal region (Figure 6E,F).

These skin sections were scored individually by histopathological analysis in a blinded manner. Each parameter of skin lesions, such as epidermal hyperplasia (Figure 7A), dermal inflammation (Figure 7B), pustule formation (Figure 7C), and rete ridge formation (Figure 7D), were analyzed. The scores of each of these lesions were combined to generate a total lesion score (Figure 7E). Vehicle control (VC) animals displayed insignificant changes compared to NC animals; however, histopathological scores of all parameters in TPA control were highly significant (*p* < 0.001) as compared to VC animals (Figure 7A–E). A decrease in epidermal hyperplasia after treatment with DEXA or WSSO 410 mg/kg was not significant; however, the higher dose of WSSO (1230 mg/kg) induced a significant decrease (*p* < 0.001) (Figure 7A) in epidermal hyperplasia.

A significant reduction in dermal inflammation was observed in all three treatment groups (*p* < 0.001). To our surprise, pustule formation and rete ridge formation (Figure 7C,D) were almost nullified in all three treatment groups, with high significance (*p* < 0.001), except for the 410 mg/kg WSSO group for rete ridge formation (*p* < 0.01) (Figure 7D). These results indicate that the topical application of WSSO (20 μL) combined with oral dose (1230 mg/kg) exhibit a highly significant reduction (*p* < 0.001) in all the lesion histopathology parameters (hyperplasia, epidermal inflammation, pustule formation, and rete ridge formation). A lower dose of WSSO (410 mg/kg) was also similarly effective except for rete ridge formation (*p* < 0.01) and hyperplasia, as compared to TPA CON (Figure 7A–D). Tropical treatment of DEXA also significantly (*p* < 0.001) reduced the observed individual lesion score parameters (Figure 7B–D) except hyperplasia (Figure 7A). Taken together, these results demonstrate significant decrease (*p* < 0.001) in the total lesion score in WSSO-treated psoriatic mouse (410 and 1230 mg/kg) compared to TPA CON animals (Figure 7E). Similarly, a significant decrease in the overall inflammation and lesion scores was observed in DEXA (*p* < 0.001) treated TPA-stimulated mice (Figure 7E).

### 3.4. In Vitro Anti-Inflammatory Activity of WSSO

#### 3.4.1. WSSO Inhibits Psoriatic Inflammation in Human Epidermoid Cells

The human epidermoid cell line, A431 was stimulated with TPA or with LPS, causing a significant increase in the release of pro-inflammatory cytokines IL-6 and TNF-α. Induction with TPA correlated with the in vivo observations and psoriatic modulation of the cytokines (Figure 8A,B), whereas LPS stimulation measured direct activation of the inflammatory pathway in epidermoid cells (Figure 8C,D). 

After pretreatment with different concentrations of WSSO, TPA- or LPS-induced secretion of cytokine expression was measured by ELISA. WSSO inhibited the stimulated release of IL-6 (Figure 8A,C) and TNF-α (Figure 8B,D) in a concentration-dependent manner. Significant (*p* < 0.001) up-regulation in the expression of IL-6 (6.3 ± 0.8 pg/mL) and TNF-α (114.4 ± 3.3 pg/mL) by TPA was observed, as compared to non-treated (NC) A431 cells (1.4 ± 0.3 and 27.4 ± 3.4 pg/mL for IL-6 and TNF-α, respectively). In TPA-stimulated A431 cells, treatment with WSSO (1 and 3 μL/mL), significantly reduced IL-6 release (Figure 8A) to 2.3 ± 0.3 and 1.8 ± 0.2 pg/mL (*p* < 0.0001). Similarly, expression of TNF-α (Figure 8B) after TPA stimulation was significantly (*p* < 0.0001) inhibited in all the WSSO treatments tested (0.1, 0.3, 1, 3, 10 μL/mL). Similarly, post LPS stimulation in A431 cells, concentration-dependent inhibition of IL-6 (Figure 8C), and TNF-α (Figure 8D) expression by WSSO was measured. Inhibition of IL-6 expression was significant (*p* < 0.05) at WSSO doses of 3–10 μL/mL (Figure 8C). WSSO also inhibited LPS-induced TNF-α expression at 3 and 10 μL/mL significantly (Figure 8D).

#### 3.4.2. WSSO Inhibits Pro-Inflammatory Cytokines in LPS-Induced THP-1 Cells

The important aspect of psoriasis is dysregulated cross-talk between immune cells responsible for inflammation control. The role of WSSO on the modulation of pro-inflammatory mediators was studied using differentiated human monocytic cells, THP-1. The effect of WSSO on THP-1 cell viability was tested (Figure 9A). At higher concentrations of 15 and 30 µl/mL, WSSO reduction in cell viability was observed. However, at the concentrations up to 10 μL/mL, changes in cell viability were insignificant and were close to untreated cells (Figure 9A). Therefore, for all the in vitro experiments, 10 μL/mL WSSO was taken as the maximum non-toxic test dose. A similar maximum non-toxic dose of 10 μL/mL of WSSO was observed for cell lines A431 and RAW264.7 (data not shown).

LPS-stimulated RAW264.7 cells exhibited highly significant upregulation in total reactive nitrogen species (RNS) production (*p* < 0.0001) (Figure 9B). RAW264.7 cells pretreated with WSSO had a significant reduction (*p* < 0.0001) in RNS compared to LPS stimulated cells at 3–10 μL/mL concentration (Figure 9B). Total RNS in control conditions (LPS alone) was taken as 100% and percentage change in RNS was calculated. 

LPS stimulation caused a significant increase (*p* < 0.0001) in cytokines IL-6 (Figure 9C) and TNF-α (Figure 9D) release from THP-1 cells. Like A431 cells, WSSO treatment significantly inhibited this elevated IL-6 expression at 3 and 10 µL/mL concentrations (*p* < 0.023 and 0.0001, respectively). A highly robust reduction in TNF-α release was observed with WSSO treatment in THP-1 cells. It was highly significant (*p* < 0.0001) at all the concentrations tested, even at 0.1 µL/mL of WSSO (Figure 9D). 

The effect of WSSO on NFκB expression was measured by luciferase reporter assay in THP-1 cells. Transiently transfected THP-1 cells were stimulated with LPS, a significant increase in NFκB expression (*p* < 0.001) was observed as compared to untransfected cells (NC) (Figure 9E). Cells pretreated with WSSO displayed concentration-dependent inhibition in the activity of the NFκB promoter. Although there was decrease in NFκB expression at 1 and 3 µl/mL, significant decrease (*p* < 0.04) was observed at 10 µL/mL of WSSO (Figure 9E).

## 4. Discussion

*W. somnifera* (WS), the Indian medicinal wonder plant, has been used since time immemorial. Traditional medicinal systems extensively used WS roots or other parts for herbal formulations. It is used in Ayurveda (called *Ashwagandha*), Chinese (called *PinYin*), Tibetan (called *Ba-dzi-gandha*), Sinhalese (called *Amukkara*), Unani (called *Asgand*), and Persian (called *Bahman*) medicinal systems. Detailed scientific research on WS medicinal values has been well described for its anti-inflammatory properties [6,46], neuroprotective [10,47], anti-cancer [48], and anti-angiogenic [49] properties. The WS plant is rich with different pharmaceutically active molecules, identified in root or leaf extracts. Surprisingly, all the biological research work on WS till now have been exclusively centered around the root extracts or leaf extracts [14,50], as root or leaf parts are suggested to have medicinal values in traditional medicinal systems. Commonly known WS oil or Ashwagandha oil is actually a cocktail of essential oils, distilled from alcoholic extracts of WS root or leaves. For some reason, the medicinal properties of the seeds of WS plant have not been investigated. Chemical composition of WS seed oil extracted by the chloroform/methanol method has been analyzed for fatty acids earlier [16]; however, the green extraction method approach by SCFE provided better yields (13%) compared to the earlier reported method (4% to 11.23%). To our knowledge, no attempts have been made to study the detailed biological activity of fatty acids extracted from WS seeds. We report the fatty acids extraction from WS seeds by SCFE, its chemical profiling, and in-depth biological activities for the first time. 

To extract fatty acids from WS seeds, SCFE technology was used, which is fairly recent technique for oily extractions. Compared to the solvent-based system of extract preparation from plant material, SCFE provides quality extracts without variation due to elegant extraction processes. In SCFE methods, highly pressurized liquid CO_2_ attains supercritical phase, penetrates the dry plant material and extracts targeted molecules [51,52]. After the vaporization of solvent CO_2_, contamination-free pure extracts can be easily recovered. CO_2_ being inert, non-toxic, inexpensive, and environment friendly is now becoming the method of choice to extract oil-like substances from a variety of oilseeds and plant materials [53]. The percentage yield of fatty acids rich oily material extracted from *W. somnifera* seeds (WSSO) was rather efficient (~13%), and was free of any solvent contaminants. 

GC-FID analysis of the oily materials provides decent insights into its chemical composition (Figure 2). In WSSO, a high amount of linoleic acid (54.15%) was identified, followed by oleic acid (23.16%) and palmitic acid (12.64%), other fatty acids such as stearic acid and nervonic acid were also identified. These fatty acids are known to be a structural part of the human epidermis [54]. A major part of WSSO, linoleic acid is an essential fatty acid known to play a beneficial role in atopic dermatitis [18], psoriasis [42], by regulating NFkB activity and inflammation [55]. Linoleic acid (LA) can be catabolized into other forms such as γ-linolenic acid and arachidonic acid. These fatty acids improve skin structure (being part of the stratum corneum and external barrier), combined with enhanced immunological properties through the inhibition of pro-inflammatory molecules, cytokines, and reactive species (ROS and RNS) [56]. Earlier studies have demonstrated that LA also causes elevation in activator protein-1 (AP-1), a transcription factor responsible for the proliferation of keratinocytes helping in skin lesion repair [55]. 11,14,17-eicosatrienoic acid (ETA) is another omega-3 polyunsaturated acid and biologically active lipid mediator [56,57] that plays an important role in wound healing and keratinocyte health [58] by inhibiting matrix metalloproteinase (MMP)-1 expression [54]. Most of the fatty acids nurture the skin tissue by helping to restore the skin’s natural barrier function and induce skin repair [59]. Such skin nurturing activities of palmtic acid [60] and oleic acid [33,61] have been well documented. Based on the chemical analysis of WSSO, its therapeutic potential against psoriasis was explored. In psoriasis pathogenesis, the primary barrier skin is broken due to enhanced inflammation. Active lipid recruitment is required to maintain structural lipid integrity; therefore, it is conceivable that WSSO components may repair psoriatic skin by supplementing some of the essential structural components. 

HPLC profiles of WSSO and the post-SCFE exhaust of WS seeds clearly show that WSSO contains trace amounts of withanolides, whereas WS seed exhaust shows good concentrations of withanolides. This observation is in line with withanolides being polar in nature and therefore non-extractable by SCFE methods. These parallel analyses indicate that the observed biological activities of WSSO are largely due to identified fatty acid contents, and not driven by withanolides.

TPA application on mouse ear skin induces psoriasis-like skin disease and is an established animal model [62]. The psoriatic model has been well characterized by skin inflammation and ear edema. After treatment with WSSO, inflammation-induced edema subsided significantly (*p* < 0.001), in a dose-dependent manner. Psoriatic lesions (hyperplasia, pustule, rete ridge, and hyperkeratosis) have been well characterized in histopathological analyses of animal skin [23]. After concurrent WSSO treatment, a robust and highly significant (*p* < 0.001) decrease in dermal inflammation, pustule and rete ridge formation was observed by an unbiased blinded scoring system. Significant inhibition (*p* < 0.001) in lesion scores was established at two different doses of 410 and 1230 mg/kg of WSSO. Therefore, WSSO displayed high anti-psoriatic-like potential by repairing psoriatic lesions and inflammation-induced edema. No abnormal changes in the animal feeding habits or changes in body weights were observed in any of the animal groups (data not shown).

Psoriatic lesions are characterized by the presence of inflammatory cells (Figure 6B), such as macrophages and neutrophils, in all the layers of the epidermis including the stratum corneum. Neutrophils have an abundance of myeloperoxidase (MPO) enzyme, with almost 5% of total neutrophil proteins. Therefore, an increment in MPO levels marks well with neutrophil infiltration in the psoriatic lesions, with a positive correlation with psoriasis severity [63]. In our study, a significant (*p* < 0.001) increase in MPO activity was observed in TPA-induced psoriatic skin biopsies, validating the study model. WSSO treatment reduced MPO activity in a dose-dependent manner with significant inhibition (*p* < 0.01) at a higher dose of 1230 mg/kg. Inhibition of MPO activity confirms a decrease in neutrophils infiltration in the epidermal region with a subsequent reduction in the severity of psoriatic lesions; this correlates well with histopathological observations.

To correlate and decipher mechanism behind the anti-inflammation and anti-psoriatic properties of WSSO, in vitro investigations were conducted using mouse (RAW264.7) and human (A431, THP-1) cell lines. Although skin cells are known to express cytokines [64], our study, however, is the first one to report TPA-induced changes in pro-inflammatory cytokines expression in the human epidermoid cell line A431, resulting in a psoriatic-like inflammation. Cells treated with WSSO displayed a marked reduction in IL-6 and TNF-α release in a concentration-dependent manner. These results correlate well with LPS-induced inflammation in A431 cells. WSSO treatment also significantly inhibited LPS-induced release of pro-inflammatory cytokines IL-6 and TNF-α in A431 cells. 

Nitric oxide (NO) is a signaling molecule playing a central role in inflammation and psoriasis [65]. Concentration-dependent reduction of RNS was observed with WSSO treatment. The anti-inflammatory potential of WSSO was also exhibited by a marked decrease in expression of pro-inflammatory cytokines, IL-6 and TNF-α. Surprisingly, a highly significant decrease in TNF-α expression was observed even at low concentrations (0.1 µg/mL) of WSSO. 

Although we have used a mouse model to study psoriasis-like disease, it is clinically reported that patients with active psoriasis have elevated levels of TNF-α in blood serum and skin lesions [34]. Currently, TNF inhibitors are used for psoriasis therapy [66]; however, due to substantially high cost and risk associated with TNF-α inhibitors [66,67], these are not commonly used. However, WSSO displays natural anti-inflammatory and anti-psoriatic properties by acute and robust inhibition of TNF-α expression. 

NFκB is known to play a key role in both psoriasis [20,34] and inflammation [35]. It is the master mediator for pro-inflammatory gene expression and functions such as the release of IL-6 and TNF-α Hence, in our study, reduction in expression of NFκB along with the downstream-associated pro-inflammatory cytokines in two different human cell lines correlate well with the mode of action of WSSO in attenuating psoriasis-like skin inflammation.

As per recommendation by the Food and Nutrition Board of the US Institute of Medicine, daily intake for omega fatty acid is safe at 12–17 g per day for adults [68], therefore the predicted 2–6 g/day consumption of WSSO would be within the recommended human doses. 

## 5. Conclusions

Here we presented, for the first time, an SCFE extraction of fatty acids from *W. somnifera* (WS) seeds along with their chemical and biological evaluations. GC-FID analysis confirmed the presence of beneficial fatty acids in WSSO, and HPLC analysis showed that WSSO contains withanolides in trace amounts. The concurrent treatment of WS seed fatty acids reduced psoriatic lesions and skin inflammation in a TPA-induced mouse psoriatic-like model. TPA- or LPS-induced cell-based assays demonstrated that WSSO indeed has strong anti-inflammatory properties in modulating NFκB activity, and in attenuating the release of pro-inflammatory cytokines, IL-6 and TNF-α. With the combination of both skin repair and anti-inflammatory properties, herbally-sourced *W. somnifera* seed fatty acids display strong anti-psoriatic potential. Our data suggest that oils isolated from *W. somnifera* seeds could well be seen as a natural remedial alternative or complimentary treatment for psoriasis-like skin inflammation. Detailed clinical studies are recommended to validate its pharmacological efficacies. 

## Figures and Tables

**Figure 1 biomolecules-10-00185-f001:**
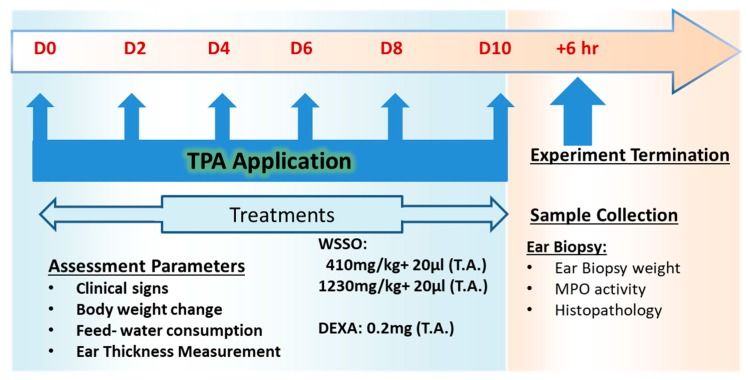
In vivo study plan to assess efficacy of *W. somnifera* seeds extracted for fatty acids (WSSO) on a 12-O tetradecanoyl phorbol 13-acetate (TPA)-induced psoriasis-like mice model: Study was started on day 0 with topical application of TPA on C57bl/6 mouse ears, with repeat applications on days 2, 4, 6, 8, and 10. Treatment with WSSO or vehicle or dexamethasone (DEXA) was given on day 1 till the termination of the experiment. During the experiment, different parameters were assessed, and followed by collection of ear biopsies on termination of the experiment. Biochemical and histopathological analysis were conducted using ear biopsies.

**Figure 2 biomolecules-10-00185-f002:**
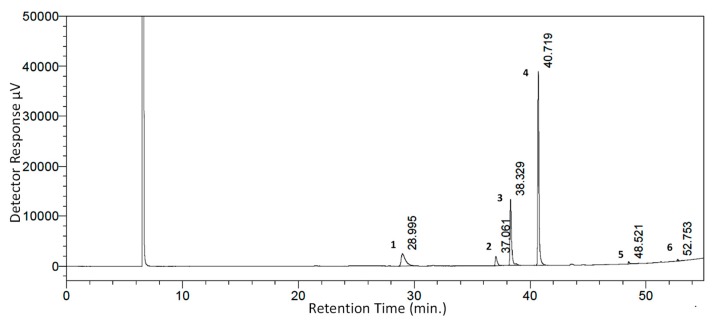
GC-FID chromatogram of WSSO: Fatty acid peaks were identified in WSSO based on different retention times. The % contents were quantified using a reference standard kit, and are listed in Table 1, as per peak numbers. Highest detector response was observed and quantified for peak 4 (linoleic acid, 54.15%), followed by peak 3 (oleic acid, 23.16%).

**Figure 3 biomolecules-10-00185-f003:**
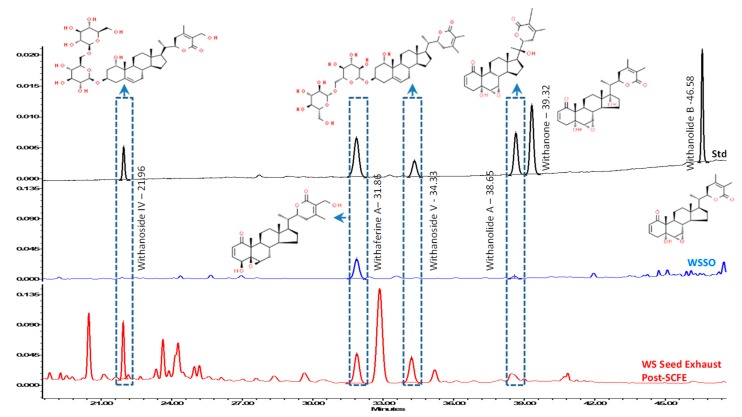
Stacked-up HPLC profiles of WSSO and the post-supercritical fluid extraction (SCFE) exhaust of WS seeds, with known standards: Withanolides contents were identified using standards mix (black line). Post-SCFE exhaust of *Withania somnifera* Dunal (WS) seeds (red line) showed good presence of withanoside IV, withaferine A, withanoside V, and withanolide A, at standard retention times (Rt). WSSO (blue line) chromatogram show identifiable peaks of withferine A (at Rt, 31.86 min) and of withanolide A (at Rt, 38.65 min). Most of the withanolides tested were identified in WS seed exhaust only. However, withanolide B and withanone were not decteced (ND) in any of the test samples. Relevent structures of withanolides are displayed at their identfied peak, as per standard mix. The observed concentrations of withanolides have been listed as % *w*/*w* with their respective retention time (Rt) in both test samples.

**Figure 4 biomolecules-10-00185-f004:**
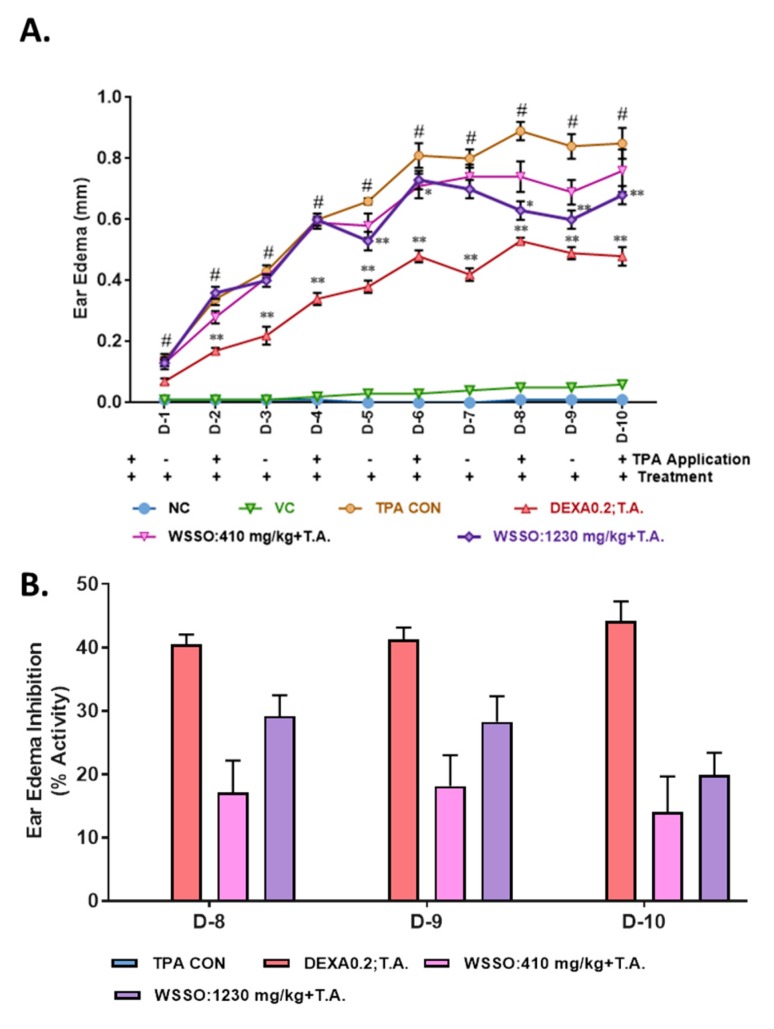
WSSO reduced ear edema in a TPA-induced psoriasis-like mouse model: Application of TPA resulted in increased ear edema, (**A**) reduction in psoriasis-induced edema was observed in WSSO treatment of 410 and 1230 mg/kg, supplemented with topical application (T.A.) of 20 µL of WSSO. At 1230 mg/kg dose, significant reduction in ear edema was observed. Dexamethasone (DEXA) was taken as positive control and acetone was used as vehicle control. (**B**) Percent ear edema inhibition was calculated by deriving data from panel (**A**), WSSO at 1230 mg/kg displayed increased inhibition of edema (20–30%) compared to 410 mg/kg (12–25%), DEXA displayed ~40% inhibition in the ear edema. Highest percentage ear edema inhibition (25–30%) was observed on day 9, with dose-dependent inhibition on the days 8, 9, and 10. All values are presented as mean ± SEM, *n* = 6 animals in each group, *p*-values, # *p* < 0.001 (NC vs. TPA), * *p* < 0.05, ** *p* < 0.001 (WSSO treated vs. TPA CON).

**Figure 5 biomolecules-10-00185-f005:**
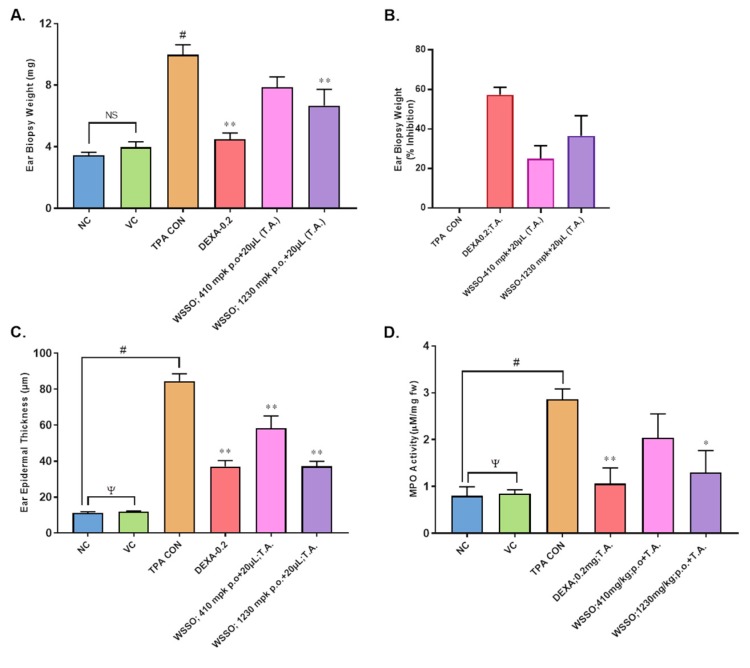
WSSO decreases TPA-induced inflammation and myeloperoxidase (MPO) activity in the ear biopsy: Dexamethasone (DEXA) was used as positive control, acetone was used as vehicle control (VC). (**A**) Reduction in ear biopsy weight was observed after treatment with WSSO, significant changes were observed in 1230 mg/kg treated mouse ears. Oral treatments were supplemented with topical application (T.A.) of 20 µL WSSO. (**B**) Higher inhibition ear biopsy weight was observed in the 1230 mg/kg treatment group (40%). (**C**) TPA induced an increase in the ear epidermis thickness that was reduced by WSSO treatment. Significant reduction in the thickness was observed with both 410 and 1230 mg/kg of WSSO. Activity of MPO was tested in ear biopsies (**D**), significant inhibition (*p* < 0.04) in MPO activity at 1230 mg/kg dose was observed compared to TPA CON. Values are presented as mean ± SEM, *n* = 6 samples in each group (*p*-values, ψ = non-significant (NC vs. VC), # *p* = 0.001 (NC vs. TPA CON), * *p* < 0.05, ** *p* < 0.001 (treated vs. TPA CON).

**Figure 6 biomolecules-10-00185-f006:**
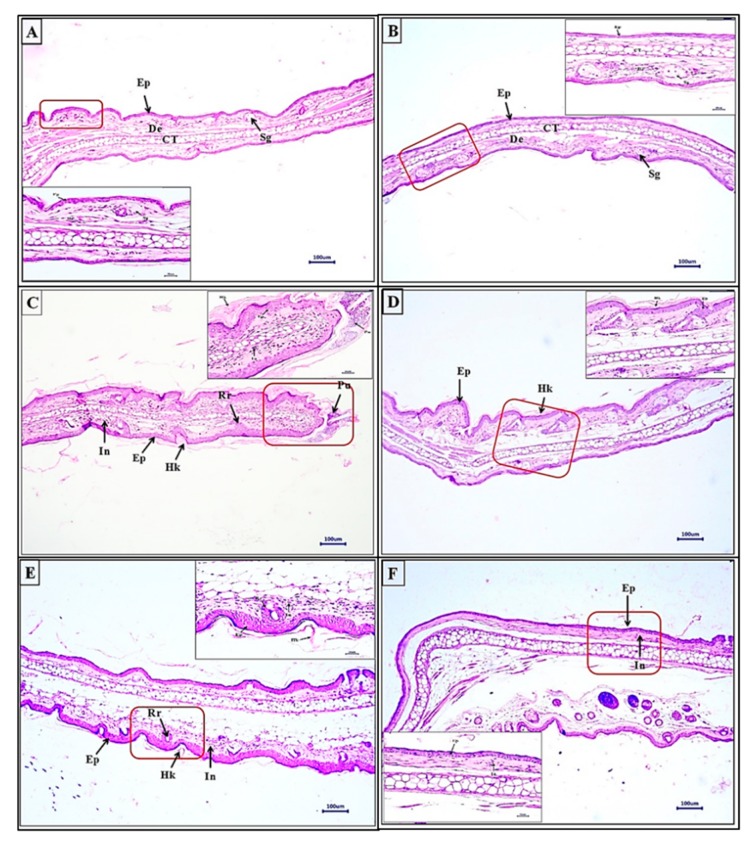
Histopathological analysis of ear biopsy from TPA-induced psoriasis-like mice model: Section of ear tissue samples were stained with H&E, (**A**) Negative control displays normal skin similar to vehicle control (**B**) with normal epidermis (Ep), dermis (De), sebaceous gland (Sg), cartilage (CT). In ear tissue of (**C**) TPA-induced psoriasis, prominent changes were observed with the presence of hyperkeratosis (Hk) and hyperplastic epidermis (Ep), inflammatory cells (In), with rete ridge (Rr) and pustule (Pu) formation. Decrease in both number and severity of these hallmarks were observed with 410 mg/kg + 20 μL T.A. (**E**) and 1230 mg/kg + 20 μL T.A. (**F**) of WSSO treatments. Dexamethasone (**D**) also significantly reduced psoriasis-like symptoms. Higher magnification images (inset) are displayed, scale bar represents 100 and 20 μm for low and high magnification images.

**Figure 7 biomolecules-10-00185-f007:**
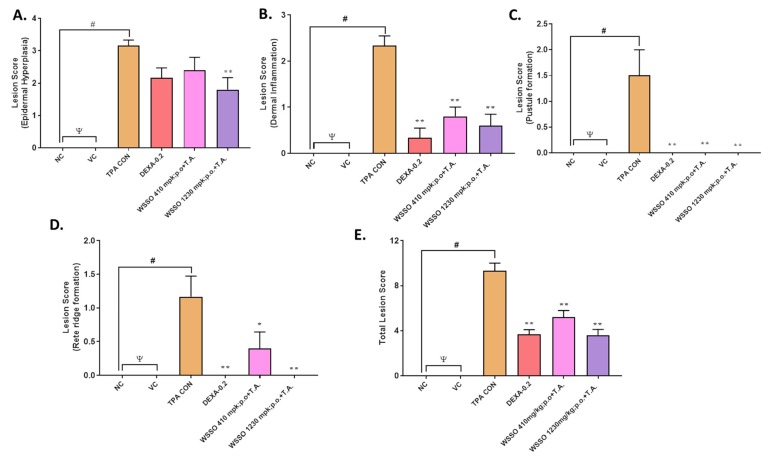
WSSO reduces the TPA-induced psoriatic histopathological score: TPA-induced inflammatory lesions were reduced by treatment with WSSO or DEXA. (**A**) Lesion score of epidermal hyperplasia reduced significantly by 1230 mg/kg WSSO treatment (*p* < 0.005). (**B**) Dermal inflammation reduced significantly post-treatment with WSSO (*p* < 0.001) at different doses in comparison to TPA CON. (**C**) Pustule formation score at epidermis reduced drastically with high significance after WSSO (*p* < 0.001) and DEX (*p* < 0.001) treatment compared to TPA CON; (**D**) Rete ridge formation was also inhibited, more significantly at 1230 mg/kg (*p* < 0.001) compared to 410 mg/kg (*p* < 0.05) of WSSO. Both of the WSSO treatments were supplemented with topical application (T.A.) of 20 µL of WSSO. (**E**) Total lesion score was calculated and significant reduction was observed after treatment with WSSO 410 and 1230 mg/kg (*p* < 0.001) and DEXA (*p* < 0.001) compared to TPA control. Values are presented as mean ± SEM, *n* = 6 samples in each group, ψ = non-significant (NC vs. VC), # *p* < 0.001 (NC vs. TPA), * *p* < 0.01, ** *p* < 0.001 (treated vs. TPA CON).

**Figure 8 biomolecules-10-00185-f008:**
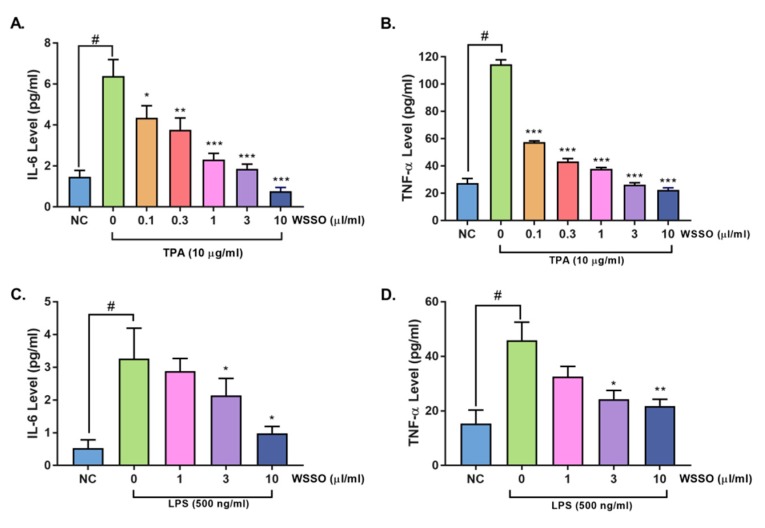
WSSO reduced pro-inflammatory cytokines release in TPA- or lipo poly-saccharide (LPS)-stimulated human epidermoid (A431) cells: A431 cells were induced with TPA (**A**,**B**), cells pre-treated with WSSO displayed highly significant (*p* < 0.0001) reduction in IL-6 release at higher concentrations of 1, 3, and 10 μL/mL, whereas inhibition of TNF-α release was highly significant (*p* < 0.0001) even at 0.1 μL/mL of WSSO. LPS stimulation increased IL-6 and TNF-α expression in A431 cells (**C**,**D**). Treatment with WSSO inhibited IL-6 (C) and TNF-α (D) expression significantly at 3–10 μL/mL concentrations. Values are presented as mean ± SEM, *n* = 3 independent experiments, *p*-values # *p* < 0.001 (LPS/TPA alone vs. non-treated (NC) cells); * *p* < 0.05, ** *p* < 0.001, *** *p* < 0.0001 (LPS/TPA alone vs. LPS/TPA + WSSO-treated cells).

**Figure 9 biomolecules-10-00185-f009:**
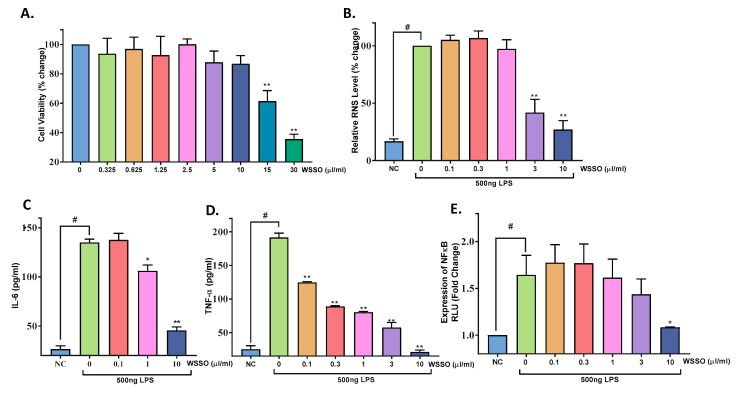
WSSO reduced total RNS in RAW264.7 cells and inhibits pro-inflammatory cytokines secretion by modulating the NFκB pathway in THP-1 cells: (**A**) Effect of WSSO on THP-1 cell viability was determined by MTT assay; loss in cell viability was observed at 15 μL/mL and above concentrations. (**B**) Reactive nitrogen species was measured in LPS-stimulated RAW 264.7 cells after treatment with non-toxic concentrations of WSSO. A significant reduction (*p* < 0.001) in RNS level (~80%) was observed at 10 μL/mL. LPS induced upregulation of cytokine IL-6 (**C**) and TNF-α (**D**) in THP-1 cells, and was inhibited by treatment with WSSO. Highly significant (*p* < 0.0001) inhibition of TNF-α was observed at all the WSSO concentrations tested, whereas significant inhibition in IL-6 expression was observed at concentrations of 1–10 μL/mL. NFκB promoter luciferase reporter assay (**E**) was performed in THP-1. Concentration-dependent inhibition of NFκB activity by WSSO was significant (*p* < 0.01) at 10 μL/mL. Values are presented as mean ± SEM, *n* = 3 independent experiments, *p*-values # *p* < 0.001 (LPS-alone vs. non-treated cells); * *p* < 0.05, ** *p* < 0.0001 (LPS alone vs. LPS + WSSO-treated cells).

**Table 1 biomolecules-10-00185-t001:** Identification of major phyto-constituents in WSSO by gas chromatography-flame ionized detector (GC-FID).

Peak No.	WSSOComponent	Mol. Formula	Chemical Structure	Retention Time (min)	Content (%)
1	Palmitic Acid(Hexadecanoic acid)	C_16_H_32_O_2_	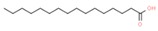	28.995	12.64
2	Stearic Acid(Octadecanoic Acid)	C_18_H_36_O_2_	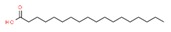	37.061	4.07
3	Oleic Acid(Cis-9-Octadecenoic Acid)	C_18_H_34_O_2_	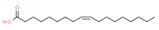	38.329	23.16
4	Linoleic Acid	C_18_H_32_O_2_	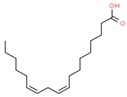	40.719	54.15
5	11,14,17-Ecosatrienoic Acid	C_20_H_34_O_2_	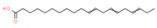	48.521	0.66
6	Nervonic Acid	C_24_H_46_O_2_	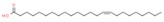	52.753	0.38

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
