# Peer review of "Super Critical Fluid Extracted Fatty Acids from Withania somnifera Seeds Repair Psoriasis-Like Skin Lesions and Attenuate Pro-Inflammatory Cytokines (TNF-α and IL-6) Release"

_biomolecules, 2020, doi:10.3390/biom10020185_

Round 1

Reviewer 1 Report

The submitted manuscript deals with supercritical carbon dioxide extraction of seeds of Withania somnifera, and with investigation of biological activity of fatty acid mixtures obtained by the extraction.

The authors of the manuscript did not consider that fact that fatty acids appear mostly as glycerides in natural sources. They can be extracted in form of glycerides by supercritical carbon dioxide procedure. In addition, natural plant oils are always accompanied by certain small quantity of free fatty acids that are more polar, however, may also be extracted with supercritical carbon dioxide in small quantities, as well as other types of plant products, e.g., withanolides mentioned in the theoretical part.

The manuscript does not contain any item of information on a way of liberating of free fatty acids from the extracted mixtures of natural glycerides. In turn, natural glycerides cannot be analyzed by GC because they are big molecules mostly unsuitable for a GC analysis. How the authors did deal with the problem of GC analysis? What they did analyze - only free fatty acids naturally present in this plant oil? In my opinion, the presented analysis is insufficient and should be accompanied by an independent method, e.g., HPLC or HPLC-MS, capable of solving the question of triglycerides and, potentially, of other accompanying plant products or their conjugates.

The analysis of the extracted material is insufficient and it is not clear what was subjected to the biological screening tests? It is obvious that the extracted material should contain also other biologically active constituents accompanying the natural plant oil, e.g. withanolides, known biologically active compounds produced by Withania spp. Therefore, my question is which evidence the authors have that the found biological effects are caused by the fatty acids only? Which evidence the authors have that the biological effects were not caused by other constituents being present in the extracted mixtures?

The manuscript in its present form should be rejected due to the lack of sufficient analysis of the extracted material, and therefore the lack of information what in fact was subjected to the biological screening tests? Subsequent results of the investigation of the biological effects of the extracted mixtures should be then re-evaluated in a view of new analysis of the material.

Author Response

The submitted manuscript deals with supercritical carbon dioxide extraction of seeds of Withania somnifera, and with investigation of biological activity of fatty acid mixtures obtained by the extraction.

The authors of the manuscript did not consider that fact that fatty acids appear mostly as glycerides in natural sources. They can be extracted in form of glycerides by supercritical carbon dioxide procedure. In addition, natural plant oils are always accompanied by certain small quantity of free fatty acids that are more polar, however, may also be extracted with supercritical carbon dioxide in small quantities, as well as other types of plant products, e.g., withanolides mentioned in the theoretical part.

The manuscript does not contain any item of information on a way of liberating of free fatty acids from the extracted mixtures of natural glycerides. In turn, natural glycerides cannot be analyzed by GC because they are big molecules mostly unsuitable for a GC analysis. How the authors did deal with the problem of GC analysis? What they did analyze - only free fatty acids naturally present in this plant oil? In my opinion, the presented analysis is insufficient and should be accompanied by an independent method, e.g., HPLC or HPLC-MS, capable of solving the question of triglycerides and, potentially, of other accompanying plant products or their conjugates.

>>Thank you very much for these insightful comments. We appreciate these fine thoughts and would like the put forward our responses:

WSSO is super critical CO2 extracted fatty acid-rich material of Withania somnifera seeds. For Withanoides estimations, we have performed HPLC analysis for WSSO and the post-SCFE residual exhaust of WS seeds. We have detected a negligible (100 times less) amount of total Withanolides in WSSO, whereas residual exhaust post-SCFE showed almost 2% total withanolides present (new Figure 3). This added data has been incorporated in the manuscript; and suggest that the observed biological responses are predominantly driven by fatty acids of WSSO.

The analysis of the extracted material is insufficient and it is not clear what was subjected to the biological screening tests? It is obvious that the extracted material should contain also other biologically active constituents accompanying the natural plant oil, e.g. withanolides, known biologically active compounds produced by Withania spp. Therefore, my question is which evidence the authors have that the found biological effects are caused by th fatty acids only? Which evidence the authors have that the biological effects were not caused by other constituents being present in the extracted mixtures?

The manuscript in its present form should be rejected due to the lack of sufficient analysis of the extracted material, and therefore the lack of information what in fact was subjected to the biological screening tests? Subsequent results of the investigation of the biological effects of the extracted mixtures should be then re-evaluated in a view of new analysis of the material.

>> As stated before, we analyzed the extracted fatty acids (WSSO) as well as Post-SCFE exhaust of WS seeds, on HPLC. We found that WSSO contains just 0.02% withanolides, whereas residual exhaust contains 2% withanolides (new Figure 3), which is almost total withanolides contents usually present. On this basis, we could conclude that SCFE method extracts only fatty acids with trace amount of withanolides, more than 99% of total withanolides still remain in the residual matter. Hence the observed biological activity of WSSO could be attributed to the fatty acids, not the withanolides. We thank you for your observations, which made us look deep into WSSO contents on HPLC.

This indeed has improved the analysis and we could correlate the observed biological responses better, the manuscript now reads better, Thank you again!  

Reviewer 2 Report

The manuscript “Super Critical Fluid Extracted Fatty Acids from Withania somnifera Seeds Repair Psoriasis-like Skin Lesions and Attenuate Pro-Inflammatory Cytokines (TNF-α and IL-6) Release” presents a good research focused on natural solutions for psoriasis. Considering the scope of the journal, the manuscript has to be more focused on biomolecules and it should highlight those of interest.

There are some points and minor mofifications:

The authors are talking about Withanoides. They should present their structures in the introduction.

Row 61: remove the etc. It means and others and add no information.

Row 98: use just the abbreviation

Row 131: present and use DEXA abbreviation at the first mention of this drug at row 91. Check all the manuscript and correct the abbreviations.

Row 219. Check the typing. It should be g or mg, but not gm!

Rows 219-230: The results present the composition of the Fatty Acid Contents in WSSO, not of all the components of WSSO. The title of the 3.1. section should reflect this. Please present the content of the fatty acids in WSSO.  The authors declare the FAME fraction was analyzed. Again what is the content of FAME in WSSO? And what does this fraction contain?

Row 231: The table should be written normally in word, not added as a picture. Re-write the chemical structure and present them in a better quality. I don’t think the CID is essential and I believe it can be removed from the table. The authors could add the retention time for each compound.

Row 241: just DEXA

Rows: 440 to 454 are repeating large sections from the introduction. This part can easily be remove.

Rows 454 to 531: do not repeat the results or methods here in the discussion section. Present the limitation of the study, like the fact that this is not psoriasis, but an animal model. Or the fact that the composition of the whole WSSO is not known, but only the fatty acids content.

Author Response

The manuscript “Super Critical Fluid Extracted Fatty Acids from Withania somnifera Seeds Repair Psoriasis-like Skin Lesions and Attenuate Pro-Inflammatory Cytokines (TNF-α and IL-6) Release” presents a good research focused on natural solutions for psoriasis. Considering the scope of the journal, the manuscript has to be more focused on biomolecules and it should highlight those of interest.

There are some points and minor mofifications:

The authors are talking about Withanoides. They should present their structures in the introduction.

>>Although Withanoides are known to be present in Withania somnifera solvent-based root and shoot extracts, however, only trace amount of withanolides were detected in WSSO on HPLC. Most of (99%) withanolides remained in the residual matter (data added in the manuscript). We have also added relevant withanolide structures in the new Figure 3, as advised.

Row 61: remove the etc. It means and others and add no information.

>>Manuscript has been updated as suggested.

Row 98: use just the abbreviation

>>These changes have been made and the manuscript updated, as suggested.

Row 131: present and use DEXA abbreviation at the first mention of this drug at row 91. Check all the manuscript and correct the abbreviations.

>>Manuscript updated as suggested, at several places.

Row 219. Check the typing. It should be g or mg, but not gm!

>>Our apologies for these typo errors, we have now corrected it to ‘g’ as several places, as suggested.

Rows 219-230: The results present the composition of the Fatty Acid Contents in WSSO, not of all the components of WSSO. The title of the 3.1. section should reflect this. Please present the content of the fatty acids in WSSO.  The authors declare the FAME fraction was analyzed. Again what is the content of FAME in WSSO? And what does this fraction contain?

>>FAME (Fatty Acid Methyl Ester) analysis of WSSO contains six identified peaks (GC-FID chromatogram has been now added as new Figure 2, for better clarity). In addition, WSSO has been analyzed on HPLC to show trace amounts of withanolides present, whereas post-SCFE exhaust of WS seeds showed most of the withanolides contents (new Figure 3) 

Row 231: The table should be written normally in word, not added as a picture. Re-write the chemical structure and present them in a better quality. I don’t think the CID is essential and I believe it can be removed from the table. The authors could add the retention time for each compound.

>>New Table 1, in word format, has been added with better quality images. Retention times as per GC chromatogram (new Figure 2) have been added, and CID information has been removed. 

Row 241: just DEXA

>>Manuscript has been updated as suggested

Rows: 440 to 454 are repeating large sections from the introduction. This part can easily be removed.

>>Thank you for this insight. The discussion section has now been modified, as suggested. This indeed has improved the readability of the manuscript.

Rows 454 to 531: do not repeat the results or methods here in the discussion section. Present the limitation of the study, like the fact that this is not psoriasis, but an animal model. Or the fact that the composition of the whole WSSO is not known, but only the fatty acids content.

>>Discussion has been modified as advised; limitations of the study have been added. Other contents of WSSO have been analyzed and added as new data along with its discussion.

We appreciate your comments and suggestions that have improved our manuscript considerably.

Reviewer 3 Report

Dr. Balkrishna and colleagues have prepared fatty acid-rich extract from Withania somnifera seeds, and analyzed its immunomodulatory activity in a psoriasis murine model. The authors have identified the main ingredients of W. somnifera seed extract. In the TPA-induced psoriasiform dermatitis model, they demonstrated that the W. somnifera extract reduced ear edema, alleviated histopathological changes, decreased myeloperoxidase activity, inhibited production of TNF-alpha and IL-6 and suppressed NF-kappaB activity. Based on these data, the authors suggest that oils extracted from W. somnifera seeds could be used as natural remedial alternative or complimentary treatment for psoriasis like skin inflammation. Although the manuscript presents some interesting data, my major concern is that these experiments provide very little novelty with respect to the fatty acid composition of W. somnifera and the treatment of psoriasis. The following concerns need also to be addressed:

Major points:

In Lines 450-453, the authors say: “For some reasons, medicinal properties of the seeds of WS plant have not been investigated. To our knowledge, no attempts have been made to study fatty acids extracted from WS seeds. This is for the first time, we report the fatty acids extracted from WS seeds, its chemical profiling and detailed biological activities.” However, the fatty acid composition of W. somnifera seeds has previously been published (Khanna PK, et al. Fatty acid composition of seed oil of Withania somnifera selectants. Physiol. Mol. Biol. Plants 2007, 13(1):73-76.). The W. somnifera seed extract was used as a combined modality of topical and oral treatment. It has not been discussed why this combination regimen was chosen. The doses of 410 and 1230 mg/kg W. somnifera seed extracts that were used for experiments have not been explained. How do these doses affect free fatty acid levels resulting from dietary intake? How do these doses match up with the recommended daily intakes for humans? Can exceeding the recommended doses cause side effects?

Minor points:

In the Materials and Methods section the specifications of chemicals are missing. The authors say that LPS was obtained from Sigma-Aldrich (Line 90). However, various LPS preparations are available from Sigma-Aldrich, which differ in their immunomodulatory activities. And also, the reader may face difficulties when trying to find W. somnifera seeds on the homepage of Patanjali Gramodhyog Trust (Line 98). The authors say that they have used standard pellet diet (Golden Feed, India) to feed laboratory animals (Line 120). Is this company identical with the Golden Feeds Ltd? Where can the composition and fatty acid content of the rodent diet used for the experiment be found? The sources of chemicals should be described with sufficient detail to allow others to repeat the experiments. In the Luciferase Reporter NFκB Expression Assay with THP-1 cells (Lines 199-209), the authors say that “consumed media was replaced with fresh media containing different concentrations of WSSO, except in negative and LPS alone control wells”. In the results section (Lines 429-432 and Fig. 7E) the authors say that “cells pretreated with WSSO displayed concentration dependent inhibition in the activity of NF-kappaB promoter”. THP-1 cells are suspension cells, therefore it may have occurred that WSSO-treated cells have been discarded. In support of the authors’ claim, this experiment should be repeated in a way when samples are treated exactly in the same way. The chromatogram obtained by using GC-FID analysis should be included in the manuscript. Sentence in Lines 235 and 236 cannot be comprehended. The formula used to calculate the total lesion score for Fig 5 should be included in the Materials and Methods section. The suggestion that the application of WS seed can be an alternative treatment option for psoriasis is an overstatement and does not seem to be a realistic option. Thus, the sentence in Line 541 should be corrected.

Author Response

Dr. Balkrishna and colleagues have prepared fatty acid-rich extract from Withania somnifera seeds, and analyzed its immunomodulatory activity in a psoriasis murine model. The authors have identified the main ingredients of W. somnifera seed extract. In the TPA-induced psoriasis form dermatitis model, they demonstrated that the W. somnifera extract reduced ear edema, alleviated histopathological changes, decreased myeloperoxidase activity, inhibited production of TNF-alpha and IL-6 and suppressed NF-kappaB activity. Based on these data, the authors suggest that oils extracted from W. somnifera seeds could be used as natural remedial alternative or complimentary treatment for psoriasis like skin inflammation. Although the manuscript presents some interesting data, my major concern is that these experiments provide very little novelty with respect to the fatty acid composition of W. somnifera and the treatment of psoriasis. The following concerns need also to be addressed:

Major points:

In Lines 450-453, the authors say: “For some reasons, medicinal properties of the seeds of WS plant have not been investigated. To our knowledge, no attempts have been made to study fatty acids extracted from WS seeds. This is for the first time, we report the fatty acids extracted from WS seeds, its chemical profiling and detailed biological activities.” However, the fatty acid composition of W. somnifera seeds has previously been published (Khanna PK, et al. Fatty acid composition of seed oil of Withania somnifera selectants. Physiol. Mol. Biol. Plants 2007, 13(1):73-76.). The W. somnifera seed extract was used as a combined modality of topical and oral treatment. It has not been discussed why this combination regimen was chosen. The doses of 410 and 1230 mg/kg W. somnifera seed extracts that were used for experiments have not been explained. How do these doses affect free fatty acid levels resulting from dietary intake? How do these doses match up with the recommended daily intakes for humans? Can exceeding the recommended doses cause side effects?

>> Thank you very much for the guidance, and our apologies for this hindsight. We have now cited the suggested publication (Khanna et.al., 2007), and discussed it in the context of our findings. Our presented fatty acid contents of WSSO correlate well with the range reported by Khanna et al, 2007. However, we have used SCFE method on WS seeds for the first time, and have also performed detailed biological studies of extracted fatty acids. Earlier work on steroidal constituents in the isolated WS oil has been already cited (Iguchi et. al.,2019) in the manuscript.

The combination treatment of test article for skin-related diseases have been used in herbal-based medicines (for example, Kar et al, 1990; Mehta et al, 2013), where the topical and oral route of administration helps in disease amelioration. Indeed, our observed oral dose dependence of biological response also support this notion. This has now been discussed in the revised manuscript.

For Dose selection, we took guidance from Simon et al, 2014, and from Dietary guidelines of US institute of Medicine, 2005, where fatty acid-containing oils have been used. The human predicted dose in our study 2 – 6 g/day is well within the recommended dietary intake of fatty acids.   

In addition, Zhao et al, 2017 have shown a robust safety profile of fatty acid-rich sea buckthorn oil up to 10 mL/Kg/day in rats for 90 days, suggesting rather safe attributes of fatty acids.  

Minor points:

In the Materials and Methods section the specifications of chemicals are missing. The authors say that LPS was obtained from Sigma-Aldrich (Line 90). However, various LPS preparations are available from Sigma-Aldrich, which differ in their immunomodulatory activities. And also, the reader may face difficulties when trying to find W. somnifera seeds on the homepage of Patanjali Gramodhyog Trust (Line 98).

>> Catalogue number of specific reagents, like LPS has now been provided in the updated manuscript.

>> Patanjali Gramodhyog Trust does not sell WS seeds for commercial purposes, however, they gifted those to us for research purposes.

The authors say that they have used standard pellet diet (Golden Feed, India) to feed laboratory animals (Line 120). Is this company identical with the Golden Feeds Ltd? Where can the composition and fatty acid content of the rodent diet used for the experiment be found?

>> For animal diets, we have inadvertently mentioned our earlier supplier (Golden feed Ltd)We have now updated with correct information and product code (Purina Lab Diet, St. Louis, MO, USA; product code 5L79). However, as the diet was same for all the animals, and the effect was observed only in animals orally treated with WSSO, in a dose-dependent manner. Therefore, we think fatty acid composition of feed may not be relevant and in the scope of current study objectives.

The sources of chemicals should be described with sufficient detail to allow others to repeat the experiments.

>>All the chemicals used in this study have now been updated with source along with cat#/ product codes, where required.

In the Luciferase Reporter NFκB Expression Assay with THP-1 cells (Lines 199-209), the authors say that “consumed media was replaced with fresh media containing different concentrations of WSSO, except in negative and LPS alone control wells”. In the results section (Lines 429-432 and Fig. 7E) the authors say that “cells pretreated with WSSO displayed concentration-dependent inhibition in the activity of NF-kappaB promoter”. THP-1 cells are suspension cells, therefore it may have occurred that WSSO-treated cells have been discarded. In support of the authors’ claim, this experiment should be repeated in a way when samples are treated exactly in the same way.

>>Yes, THP-1 cells are suspension cell lines, however, we used PMA induced differentiated THP-1 (macrophage), which are adherent, therefore a change of media does not cause the removal of cells. Thank you for these insights, the method section has now been updated to describe the method of THP-1 differentiation and their subsequent usages.

The chromatogram obtained by using GC-FID analysis should be included in the manuscript.

>>Chromatogram of GC-FID has now been added to the manuscript with a list of identified peaks, as new Figure 2, for better clarity and results display.

Sentence in Lines 235 and 236 cannot be comprehended. The formula used to calculate the total lesion score for Fig 5 should be included in the Materials and Methods section. The suggestion that the application of WS seed can be an alternative treatment option for psoriasis is an overstatement and does not seem to be a realistic option. Thus, the sentence in Line 541 should be corrected. 

>>Method to compute total lesion scores has now been updated in materials and Method section (As suggested, the manuscript has been modified and lines have been removed/edited to avoid overstatement, in general.

We thankful for you to go through our manuscript with such fine comb, this has helped us to enhance the overall quality of our manuscript.

Reviewer 4 Report

Dear Authors, the paper develops the theme well also from a methodological point of view, small changes must be made to the text, in particular I point out: 

correct NFKB with NF-kB line 24,37,80,81,199,200,209,290,405,413,427,428,430,431,468,527,529,538

line 36 do not use the word ingredients but compounds

line 227 replace linoleic acid with linolenic acid

line 531 remove a point

Author Response

Dear Authors, the paper develops the theme well also from a methodological point of view, small changes must be made to the text, in particular I point out: 

correct NFKB with NF-kB line 24,37,80,81,199,200,209,290,405,413,427,428,430,431,468,527,529,538

>>Thank you very much for spotting this error. We have now corrected NF-kB throughout the manuscript.

line 36 do not use the word ingredients but compounds

>>Manuscript has been updated, as suggested.

line 227 replace linoleic acid with linolenic acid

>>Thank you, the manuscript has been updated after correction, at a couple of more places as well.

line 531 remove a point

>>Manuscript modified as suggested, Thank you.

Round 2

Reviewer 1 Report

The authors improved the manuscript in revision by adding new data. The manuscript gives much more adequate responses, and the research described was conducted in adequate way now.

I recommend the manuscript for publication in its revised form.

Reviewer 3 Report

The novelty of this study has now been clearly specified. The comments have been fully addressed and the manuscript has substantially been revised.